# Artificial Neural Network for the Fast Screening of Samples from Suspected Urinary Tract Infections

**DOI:** 10.3390/antibiotics14080768

**Published:** 2025-07-30

**Authors:** Cristiano Ialongo, Marco Ciotti, Alfredo Giovannelli, Flaminia Tomassetti, Martina Pelagalli, Stefano Di Carlo, Sergio Bernardini, Massimo Pieri, Eleonora Nicolai

**Affiliations:** 1Department of Experimental Medicine, Policlinico Umberto I, ‘Sapienza’ University, 00161 Rome, Italy; cristiano.ialongo@uniroma1.it; 2Laboratory of Clinical Microbiology and Virology, Tor Vergata University Hospital, 00133 Rome, Italy; marco.ciotti@ptvonline.it; 3Department of Experimental Medicine, University of Rome Tor Vergata, 00133 Rome, Italy; alfredo.giovannelli@gmail.com (A.G.); flaminia.tomassetti@students.uniroma2.eu (F.T.); bernards@uniroma2.it (S.B.); 4Department of Industrial Engineering, University of Rome Tor Vergata, 00133 Rome, Italy; 5Department of Laboratory Medicine, Tor Vergata University Hospital, 00133 Rome, Italy; pelagallimartina90@gmail.com (M.P.); stefano.dicarlo@ptvonline.it (S.D.C.); 6Department of Biomedicine and Prevention, University of Rome Tor Vergata, 00133 Rome, Italy; 7Departmental Faculty of Medicine, Unicamillus-Saint Camillus International University of Health and Medical Sciences, 00131 Rome, Italy

**Keywords:** urinary tract infections, UTI, bacteriuria, urine culture, flow cytometry, neural networks, multilayer perceptrons

## Abstract

**Background:** Urine microbial analysis is a frequently requested test that is often associated with contamination during specimen collection or storage, which leads to false-positive diagnoses and delayed reporting. In the era of digitalization, machine learning (ML) can serve as a valuable tool to support clinical decision-making. **Methods:** This study investigates the application of a simple artificial neural network (ANN) to pre-identify negative and contaminated (false-positive) specimens. An ML model was developed using 8181 urine samples, including cytology, dipstick tests, and culture results. The dataset was randomly split 2:1 for training and testing a multilayer perceptron (MLP). Input variables with a normalized importance below 0.2 were excluded. **Results:** The final model used only microbial and either urine color or urobilinogen pigment analysis as inputs; other physical, chemical, and cellular parameters were omitted. The frequency of positive and negative specimens for bacteria was 6.9% and 89.6%, respectively. Contaminated specimens represented 3.5% of cases and were predominantly misclassified as negative by the MLP. Thus, the negative predictive value (NPV) was 96.5% and the positive predictive value (PPV) was 87.2%, leading to 0.82% of the cultures being unnecessary microbial cultures (UMC). **Conclusions:** These results suggest that the MLP is reliable for screening out negative specimens but less effective at identifying positive ones. In conclusion, ANN models can effectively support the screening of negative urine samples, detect clinically significant bacteriuria, and potentially reduce unnecessary cultures. Incorporating morphological information data could further improve the accuracy of our model and minimize false negatives.

## 1. Introduction

Urinary tract infections (UTIs) are among the most common infectious diseases encountered in hospital and community settings [1]. The term “UTI” broadly refers to the presence of bacteria in urine at concentrations exceeding 10^5^ CFU/mL in adults, a condition known as bacteriuria [2]. However, bacteriuria alone does not necessarily equate to disease. Distinguishing between asymptomatic bacteriuria and symptomatic UTI is essential to avoid unnecessary antibiotic treatment and focus medical intervention on patients with true clinical infections [3].

The diagnosis of a UTI usually involves a combination of clinical symptoms, signs, and laboratory tests, with urine culture considered the traditional gold standard. Despite its diagnostic accuracy, urine culture is time-consuming and labor-intensive, often requiring 24–48 h for results to be obtained. This delay can lead to inappropriate antibiotic use and prolonged discomfort for patients [3,4]. Moreover, routine urine culture without solid clinical suspicion risks yielding false-positive results, which strains both laboratory resources and clinical decision-making.

A more nuanced approach involves using the patient history and presenting symptoms to guide the decision to perform a urine culture. Because UTI diagnosis is primarily clinical, urine culture should serve to confirm an already well-founded suspicion, rather than to screen indiscriminately. Flow cytometry and emerging diagnostic technologies, including machine learning (ML), offer promising tools to enhance this clinical assessment, reduce unnecessary cultures, and streamline diagnosis [5,6,7,8,9,10].

Flow cytometry enables rapid quantification of bacteria and leukocytes in urine, providing valuable preliminary data that can help determine the likelihood of a UTI [5,11,12]. When incorporated into a diagnostic algorithm, these parameters can efficiently identify patients who require further microbiological analysis. This approach is particularly useful for avoiding overdiagnosis in cases of asymptomatic bacteriuria, such as in patients with indwelling catheters or urinary tract anomalies, or during pregnancy.

The traditional threshold of ≥10^5^ CFU/mL, established over 60 years ago [13,14], is increasingly questioned. Modern diagnostics emphasize a more patient-centered approach that considers symptomatology, the bacterial load, and additional urinary markers. For example, leukocyte counts may help distinguish contamination or colonization from true infection [15,16]. Therefore, flow cytometry has become a powerful tool for re-evaluating outdated criteria such as the Kass threshold, offering a more dynamic and clinically relevant diagnostic aid [4].

Artificial neural networks (ANNs) are a class of machine learning (ML) algorithms inspired by the structure and functioning of the human brain. They are particularly well-suited for classification problems, where the objective is to map a set of input features to discrete output classes [17]. ANNs learn these mappings through a process of supervised training, during which the network adjusts its internal parameters (weights and biases) based on the error between predicted and actual labels. One of the primary strengths of ANNs lies in their flexibility and adaptability. They can process both qualitative (categorical) and quantitative (numerical) data, which makes them versatile tools across a wide range of applications. ANNs can perform binary classification (distinguishing between two classes) or multiclass classification (differentiating among more than two categories). The core architecture of an ANN consists of interconnected layers of artificial neurons (so-called perceptron): an input layer, one or more hidden (integrating) layers, and an output layer. A key advantage of ANNs is their data-driven nature. Unlike rule-based systems, their performance is not constrained by pre-defined logic but is instead determined by the characteristics and quality of the training data. As more data become available, ANNs can continue to learn and improve, enhancing both their accuracy and generalizability [18].

This property is especially valuable in fields characterized by large, complex, and high-dimensional datasets. In the context of UTI diagnostics, ANNs offer the potential to automate and enhance decision-making processes by integrating diverse data sources, including laboratory test results, patient demographics, symptom profiles, and flow cytometry parameters [19].

This study aims to explore the integration of an ANN into the pre-analytical workflow for UTI diagnosis to reduce unnecessary urine cultures. Therefore, we aim to develop a model that represents a more efficient, cost-effective, and clinically sound approach to sample management and that reduces costs and prevents false positives.

## 2. Results

A total of 8181 samples were extracted from the LIS database and analyzed. Demographic characteristics of the study population are summarized in Table 1.

Table 2 presents the distribution of positive and negative results for the dataset of yeasts and bacteria, which served as the output variables for the ANN-MLP. Initial evaluations revealed that the ANN-MLP performed poorly in distinguishing yeast positivity, as urine strip tests and urinalysis provided limited diagnostic value. Consequently, yeast was excluded from further modeling, and only the presence of bacteria was used as the output variable.

The dataset chosen for the analysis comprised 18 variables, namely microbes, color, urobilinogen, nitrite, erythrocytes, aspect, bilirubin, age, ketones, proteins, specific gravity, leucocytes, acidity, epithelial cells, hemoglobin, esterases, sugar, and gender. Based on their normalized importance, only three variables—“microbes” (0.308), “color” (0.096), and “urobilinogen” (0.077)—exceeded the inclusion threshold of 0.070 (Figure 1).

Comparative modeling was performed by using all 18 variables and three simplified subsets.

As shown in Table 3, the full 18-variable model performed the best overall. While the positive predictive value (PPV) never exceeded 88%, limiting clinical utility for confirming infections, the negative predictive value (NPV) consistently exceeded 95%, making the model reliable for excluding infections.

Notably, the variable “nitrite”, typically associated with urinary tract infections, had a low importance score (0.068) and showed no significant distribution difference between culture-positive and culture-negative cases (Table 4), although it remained a valid marker for negative cultures.

The distribution of the “microbes” variable is illustrated in Figure 2. This variable is not synonymous with CFU, but reflects the number of “objects” (the microbes) that the analyzer classifies as bacteria. This count tends to be underestimated when the bacterial load is elevated in the sample, as can be observed in Figure 2.

Importantly, only the full 18-variable model was able to recognize contaminated specimens, albeit with low specificity (3%). This model resulted in 0.67% of the cultures being unnecessary cultures (18 false positives vs. 2680 true negatives).

The reduced model, based on the three most informative variables, yielded an NPV of 96.6%. Further reduction to just two inputs—“microbes” plus either “color” or “urobilinogen”—did not significantly affect the NPV or overall accuracy (Table 3). The models’ unnecessary culture rates ranged from 0.82% (22 false positives vs. 2676 specimens) to 0.63% (17 false positives vs. 2681 specimens). Given that “urobilinogen” is instrument-measured while “color” may be assessed visually, the final ANN-MLP model used “microbes” and “urobilinogen” as inputs (Figure 3), with respective absolute importances of 0.232 and 0.0768. The synaptic weights are detailed in Table 5.

In contrast, using “microbial count” alone resulted in a reduced NPV of 93.1% (95% CI: 92.2–94.0%).

## 3. Discussion

Urine analysis with reflex to culture (URTC) is a diagnostic strategy in which the results of urinalysis and urine dipstick tests are combined to determine whether a urine culture should be performed, which thereby excludes the presence of a urinary tract infection (UTI) while avoiding unnecessary cultures [20,21,22].

Traditionally, URTC is based on the linear combination of various parameters to produce a score or apply multi-parametric cutoffs. This approach assumes that there is a sharp distinction between cases, and that relevant variables shift predictably across diagnostic groups. However, biological variability, pre-analytical inconsistencies, and stochastic factors often obscure these patterns, challenging the effectiveness of purely linear models [22].

In the present study, we adopted a non-linear approach by implementing a multi-layer perceptron (MLP) to evaluate the discriminatory value of routine urinalysis variables. Historically, MLPs have been used to recognize complex, often blurred patterns, such as those in image or signal processing [23]. Importantly, MLPs are relatively simple to implement and can be integrated into laboratory information systems (LIS) or middleware, offering intelligent support not only in diagnostic decision-making but also in pre-analytical sample management [24,25].

Our findings show that an MLP model that uses only two input variables—“microbes” and “urobilinogen”—achieves an NPV exceeding 95%, implying that less than 1% of negative samples are unnecessarily sent for culture. This is a promising result for URTC workflows, as it helps to reduce both unnecessary laboratory work and the risk of false positives, particularly in the absence of clear clinical indications for infection. Such performance demonstrates the potential utility of a minimal, low-complexity ANN-based system for efficient rule-out strategies.

Nonetheless, it is necessary to acknowledge the limitations of the model. Even when all 18 available variables are included, the MLP’s PPV does not exceed 88.3%. This means that, while the model is good at excluding infection, it is less reliable for confirming it, with a true-positive/false-positive ratio of less than 2:1. This limitation is further highlighted by the F1 score, which integrates the PPV and NPV and never surpasses 90%, which points to a limited overall classification performance when it comes to true UTI confirmation.

These findings suggest some important considerations for both improving the performance of MLPs and re-evaluating the diagnostic value of traditional urine test variables [26]. To correctly identify UTI-positive samples, two types of signals should ideally be present: one indicating microbial presence (e.g., bacteria) and another indicating the host’s immune response to infection. The presence of one without the other can be misleading: isolated bacterial detection may indicate contamination, while markers of inflammation alone may result from non-infectious conditions. This rationale explains why a simple model including two non-redundant variables can match the performance of more complex models.

The “microbes” variable, derived from flow cytometry, is the most informative input and acts as a proxy for bacterial load. This is not surprising, as it reflects the “burden” of microbial presence in the sample. However, as shown in Figure 2, microbial counts overlap among contaminated, negative, and positive samples. Such overlap can be attributed to variations in the infection stage at the time of collection, and potentially to saturation effects inherent in the cytometric method itself. Indeed, the microbial counts in our dataset never exceeded 98,000, even though the actual bacterial loads may have been higher. This artificial ceiling may flatten variability and mask true differences between groups. One possible corrective measure is to flag and dilute high-count samples for reanalysis to thereby enhance the accuracy of microbial quantification, independently of MLP use.

Regarding inflammatory markers, our model surprisingly found minimal predictive value in variables that are traditionally associated with UTI, such as leukocytes, epithelial cells, nitrites, and esterase. Post-hoc simulations (data available on request) confirmed this finding: combining “leukocytes” and “nitrites” yielded an F1 score of 81.3%, while “leukocytes” and “esterase” dropped the score further to 57.7%. Table 3 corroborates this, showing that less than 1.5% of the culture-negative samples were nitrite-positive (out of 2698 cases), and that the nitrite distribution was essentially equivalent between the culture-positive and culture-negative samples. While nitrites still demonstrate a high NPV, alone or with esterase [27,28], their low PPV limits their utility in URTC workflows. In our model, “microbes” appear to complement nitrite testing, not the other way around.

Another intriguing and unexpected result was the contribution of the urine “color” or its biochemical surrogate, “urobilinogen,” to the model’s NPV. Currently, there is no literature directly supporting a link between these parameters and bacterial absence. This may represent a spurious correlation, due to the predominance of bacteria-negative samples in the dataset. As a result, the MLP may be learning an indirect association: these parameters frequently appear in negative cases and are therefore interpreted as indicators of negativity, enhancing the NPV. This type of aberration could potentially indicate model overfitting—a common issue where an MLP learns patterns from noise in the given data. In our case, however, the 95% CIs of the performance metrics during training were consistently close to those observed during testing. Furthermore, as shown in Table 3, these metrics remained statistically similar even as the number of input variables was reduced from 18 to 2. Although SPSS does not support regularization, there were no indications of overfitting affecting our model. At least, this seems to further suggest that the test panel lacks a robust, inflammation-specific marker tied to microbial infection, which would balance the weight of the microbial count.

Two key limitations must be acknowledged when interpreting the results of this study. First, although the dataset includes 8181 cases, it is not large in the context of training and validating MLPs across 18 variables, particularly given that positive samples comprise less than 10% of the dataset. The resulting class imbalance (ratio > 10:1) causes the model to learn primarily to exclude UTI, which explains the strong NPV and modest PPV. Of course, the use of synthetic data resampling techniques (e.g., SMOTE or ADASYN) could help mitigate this limitation. However, based on what the variable importance shows (Figure 1), it should be considered that the real limitation in our data lies in the count of microbial particles (See Figure 2), which tends to rapidly reach saturation. Saturation is a condition of reduced variability that leads to a high probability of replicating the same information. Therefore, synthetic resampling using the k-nearest neighbors algorithm would merely create new cases within a very narrow value space, thus providing no real benefit in training the ANN. Second, the dataset lacks the direct clinical diagnoses of UTI. Thus, the model classifies samples based on their bacteriuria status rather than confirmed infection. Considering that a true diagnosis would ideally integrate bacterial presence with host response markers, the MLP is ultimately trained to detect “clinically significant bacteriuria” and likely excludes contamination, but not necessarily true UTIs. This would indeed be a kind of cross-validation that would further strengthen the soundness of our approach in pre-analytics. Future investigations will focus on multicenter studies to broaden the patient cohort and investigate different clinical settings.

## 4. Materials and Methods

### 4.1. Study Design and Algorithm Used

Samples: a total of 9599 entries were extracted from the LIS (Modulab 3.1.02, Instrumentation Laboratory S.p.A., Werfen, Milan, Italy) within the years 2020–2023, of which 8181 were used for the final analysis. The variables considered and their respective types were as follows:Patient: age (scale), gender (nominal);Urine aspect: color (nominal), aspect (nominal);Urine test strip: acidity (scale), specific gravity (scale), protein (discrete), sugar (discrete), ketones (discrete), bilirubin (discrete), urobilinogen (discrete), nitrite (dichotomic), esterases (discrete), hemoglobin (discrete);Urinalysis: leucocytes (scale), erythrocytes (scale), epithelial cells (scale), microbes (scale).

Method: artificial neural network (ANN) to integrate information from LIS to predict negative samples.

To obtain reproducible results and reduce any possible bias in the training process, the cases in the database were randomly ordered using a random number generator. The same principle was applied to define the order of the input variables (factors and covariates) through which the data were presented to the neural network. Finally, the database was partitioned 2:1 into a training group (*n* = 5271, 64.43%) and a test group (*n* = 2910, 35.57%), ensuring that the frequencies of nominal and ordinal variables agreed with the training/test ratio of 1.8 and that the medians of continuous variables did not differ significantly (Mann–Whitney test *p*-value > 0.05). Due to low frequency, a factor was represented in only one of the two groups, so the database partition was adjusted to ensure the proportion respected at least a 1:1 training/test ratio. Finally, the percentage of positive and contaminated samples corresponds to 6.70% and 3.53%, respectively, in the training subset, and 7.29% and 3.47%, respectively, in the test subset.

The ANN was built according to the multi-layer perceptron (ANN-MLP) architecture. The scale-dependent variables were rescaled by the adjusted normalized procedure. A single hidden layer of artificial neurons (with no restriction on the number) was set using a hyperbolic tangent activation function. The same function was used for the output neurons. The chosen training strategy was “online” with the gradient descent optimization algorithm. The initial learning rate was set to 10^−3^, with a lower bound of 10^−9^ and a learning rate reduction after 5 epochs.

The training objective was to maximize the NPV of urine classification in the bacterioscopic exam (negative/contaminated/positive), in order to minimize the number of input variables used.

To compute the NPV, considering the aim of avoiding unnecessary microbial cultures, “contaminated” samples that the ANN-MLP classifies as negative are added to those correctly recognized as negative, while those classified as positive are added to those incorrectly recognized as negative.

To obtain a minimal model, the process started with a complete model including all input variables, after which those with an absolute weight lower than 0.07 were removed. This process was repeated three times to verify the consistency of the final result, together with the 95% CI of the PPV and NPV.

All calculations were performed with IBM SPSS 27.0 (IBM Corp., Armonk, NY, USA).

### 4.2. Sample Collection

The samples were analyzed within 2–5 h from sampling for routine culture, assumed as the reference method, by the Sysmex UF-5000 analyzer (manufactured by Sysmex Europe, Norderstedt, Germany). The collection of specimens adhered to the European Urinalysis Guidelines, utilizing sterile disposable tubes devoid of preservatives [29]. All procedures were executed in compliance with institutional and national ethical norms, as well as the principles outlined in the Declaration of Helsinki (2008).

### 4.3. The Sysmex UF-5000 Analysis

The Sysmex UF-5000 is a fully automated analyzer that identifies and counts the formed elements in uncentrifuged urine samples by using flow cytometry and impedance methods.

This instrument contains two chambers where diluted urine is incubated with specific dyes and lysis reagents. One chamber exclusively detects bacteria, while the other counts particles such as erythrocytes, leukocytes, and casts. After staining, the samples are transported to a flow cell, where they are analyzed using a semiconductor laser and characterized by forward scatter, side scatter, and fluorescence. In this study, only the bacterial count is utilized.

The quality controls (QCs) were analyzed every morning.

### 4.4. Microbiological Analysis

The samples were directly sown on a BD CHROMagar Orientation chromogenic medium plate (Becton Dickinson GmbH Company, Heidelberg, Germany), which was non-selective to serve the purpose of isolating, directly identifying, distinguishing, and quantifying pathogens associated with urinary tract infections.

## 5. Conclusions

In conclusion, the use of a MLP offers a low-cost, easily deployable approach to support URTC strategies, especially in high-throughput laboratory settings. Nevertheless, obtaining faster results could help physicians in optimizing antibiotic therapy, which would ultimately contribute to the fight against antimicrobial resistance [30,31]. However, further refinements are needed to improve the predictive value of the model. These include enhancing its measurement precision for key variables, mitigating class imbalance in the training dataset through synthetic resampling if appropriate, and possibly integrating novel biomarkers that are more specifically linked to infection-related inflammation. Such improvements would help realize the full diagnostic potential of ANNs in the context of urine microbial analysis.

## Figures and Tables

**Figure 1 antibiotics-14-00768-f001:**
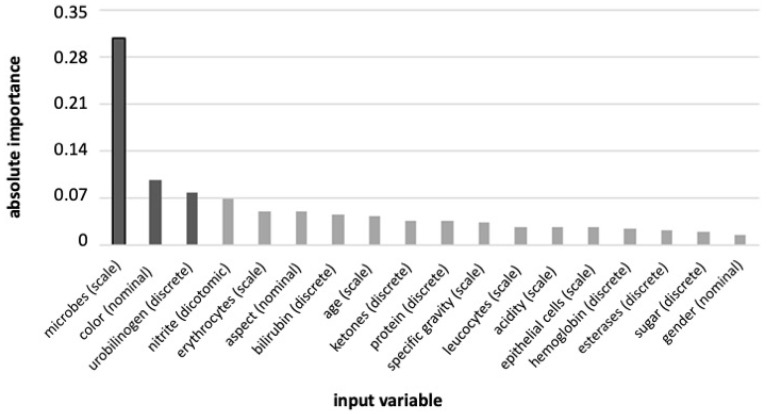
Variable importance ranking and selection for MLP model reduction. Input variables and type (scale, discrete, or nominal) in the building of the multi-layer perceptron (MLP) are ranked by their absolute importance. A cutoff of 0.07 is used afterward to select the minimum number of variables for building the reduced model based on their absolute importance.

**Figure 2 antibiotics-14-00768-f002:**
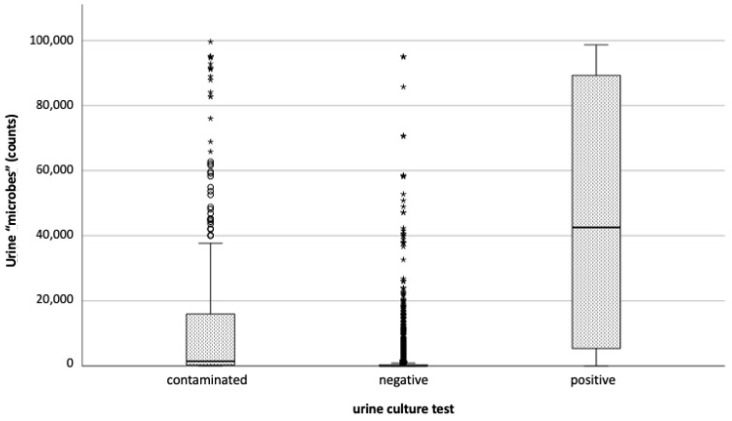
Box plot of counts of “microbes” in the specimens by the urine culture test; empty circles (°) represent extreme values, stars (*) are outliers.

**Figure 3 antibiotics-14-00768-f003:**
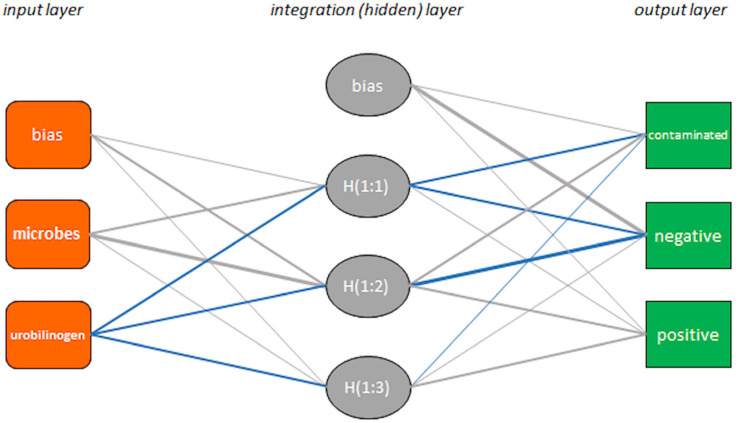
The architecture of the multi-layer perceptron (MLP) of the reduced model based on the variables “microbes” and “urobilinogen”. The thickness of the connectors is directly proportional to the stimulation weight (w), while the color of the connectors changes according to whether the input is excitatory (gray if −1 ≤ w < 0) or inhibitory (blue if 0 < w ≤ 1). Synaptic weights are provided in Table 5.

**Table 1 antibiotics-14-00768-t001:** Demographic features of the study population used to train and test the ANN-MLP.

	Males	Females	Aggregate
N	2968	5213	8181
Age (years):			
Average (SD)	57.4 (18.2)	46.4 (17.9)	50.4 (18.8)
Median (IQR)	61 (23)	43 (27)	51.0 (31)
Outpatients (%N)	94.6	96.7	96.1

**Table 2 antibiotics-14-00768-t002:** Positivity and negativity distribution regarding yeasts and microbes.

	Males	Females	Aggregate
N	2968	5213	8181
Microbes: Positives (%N)	5.5	7.7	6.9
Microbes: Negatives (%N)	92.6	87.9	89.6
Microbes: Contaminated (%N)	1.9	4.4	3.5
Yeasts: Positives (%N)	0.7	0.9	0.8
Yeasts: Negatives (%N)	99.3	99.1	99.2

**Table 3 antibiotics-14-00768-t003:** MLP model performance for predicting urine culture outcomes using different input combinations.

Variables (Inputs) in the Model	PPV%	(95%CI)	NPV%	(95%CI)	Accuracy%	F1%	MR%	UMC%
Full (18 inputs) *	88.3	(85.7 to 90.9)	97.2	(96.9 to 97.6)	96.8	86.3	3.2	0.67
microbes + color + urobilinogen	86.9	(84 to 89.7)	96.6	(96.3 to 97)	96.2	83.1	3.8	0.67
microbes + color	84.6	(81.6 to 87.6)	96.7	(96.4 to 97.1)	96.1	83.1	3.9	0.63
microbes + urobilinogen	87.2	(84.3 to 90.1)	96.5	(96.2 to 96.9)	96.1	82.6	3.9	0.82

* = see variables presented in Figure 1; PPV = positive predictive value; NPV = negative predictive value; 95%CI = 95% probability confidence interval; F1 = average of F1 metrics; MR = misclassification rate; UMC = unnecessary microbial cultures.

**Table 4 antibiotics-14-00768-t004:** Distribution of the presence of nitrite in the urine specimens by the urine culture test result (N = 8181).

		Nitrite	
		Absent (%N)	Present (%N)
Urine culture test	Contaminated	2.87	0.64
	Negative	89.08	0.50
	Positive	3.37	3.53

**Table 5 antibiotics-14-00768-t005:** Synaptic weights (parameter estimation) of the multi-layer perceptron (MLP) of Figure 3 based on use of “microbes” and “urobilinogen” inputs to predict urine culture test outcome (contaminated/negative/positive).

		Hidden Layer (Integration)			Output Layer		
**Input layer**		H (1:1)	H (1:2)	H (1:3)			
	bias	0.042	0.821	0.107			
	microbes	0.780	1.477	0.11			
	urobilinogen	−0.327	−0.319	−0.141			
					contaminated	negative	positive
**Hidden layer** **(integration)**	bias				0.010	1.208	0.140
	H (1:1)				−0.483	−0.43	0.029
	H (1:2)				0.504	−0.93	0.535
	H (1:3)				−0.116	0.129	0.410

## Data Availability

Data is contained within the article.

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
