# Peer review of "Artificial Neural Network for the Fast Screening of Samples from Suspected Urinary Tract Infections"

_antibiotics, 2025, doi:10.3390/antibiotics14080768_

Round 1

Reviewer 1 Report

Comments and Suggestions for Authors

This manuscript, entitled “Artificial Neural Network in the fast screening of samples from suspected urinary tract infections,” explores the use of artificial neural networks, specifically a multilayer perceptron, to assist in pre-analytical triage of urine samples for suspected urinary tract infections. The study leverages a large dataset (n = 8,181) of urine samples to evaluate whether artificial neural network models can identify negative or contaminated samples, thereby reducing unnecessary urine cultures.

The manuscript is well-structured, methodologically sound, and presents clear and reproducible results. The integration of machine learning into diagnostic workflows is timely, and this study contributes meaningfully to the field of laboratory medicine and antimicrobial stewardship. However, several limitations and areas for improvement should be considered. Thus, the paper is suitable for publication after addressing the concerns above, particularly the clarification around classification targets (bacteriuria vs. UTI), class imbalance effects, and better contextualizing variable importance. 

Suggestions: 

    • Acknowledge more clearly in the abstract and discussion that the model detects bacteriuria, not clinical UTI. Also, recommend future validation against clinical outcomes.
    • Consider SMOTE or other resampling techniques in future work, or at least simulate their effect in a supplemental analysis.
    • Clarify this further in the discussion or perform a post-hoc analysis to rule out overfitting.
    • Acknowledge this limitation more explicitly in the conclusion.
    • In the abstract section, write: “Contaminated samples represented 3.5% of cases and were predominantly misclassified as negative by the MLP.” Instead of Contaminated samples were 3.5%, and the MLP classified them as negatives.”
    • Figure 1 and Table 3 are central to the findings. Consider labeling them more clearly with meaningful titles (e.g., "Model Performance with Variable Inputs").
    • In the abstract and results, use “specimens” instead of “samples” for medical precision.
    • Clarify that “microbes” as a variable refers to flow cytometry counts, not CFU.
    • Reference [11] (Kass) is historically significant. You may also want to reference more recent critiques of the Kass threshold to support your position.
    • The conclusion is appropriate, but could benefit from a stronger emphasis on the clinical implications and next steps

Author Response

Comment 1: Acknowledge more clearly in the abstract and discussion that the model detects bacteriuria, not clinical UTI. Also, recommend future validation against clinical outcomes.

Responce 1: We added a sentence in the abstract (line 40), while in the discussion paragraph the concept expressed by the reviewer was already addressed in lines 283 to 287. A sentence was also added in the discussion (lines 288-290) about the clinical validation.

Comment 2: Consider SMOTE or other resampling techniques in future work, or at least simulate their effect in a supplemental analysis.

Responce 2: The reviewer highlights an important point. Synthetic resampling can mitigate the effects of this phenomenon; however, it does not appear to be the most suitable strategy in our case. The reason lies in the attenuation of class separation caused by the saturation of microbial particle readings. While it is possible to resample the minority populations (“positive” and “contaminated”), this merely replicates the saturation already present in the readings. In other words, there is no synthetic resampling technique based on k-nearest neighbors that can mitigate the effect of variable saturation, because saturation does not leave "empty spaces" to be filled—instead, it tends to overfill the available space. Thus, ideally, SMOTE would only further saturate what is already saturated. A brief explanation was added accordingly in the text.

Comment 3: Clarify this further in the discussion or perform a post-hoc analysis to rule out overfitting.

Response 3: The MLP show quite steady performance as we reduced inputs from 18 to 2. In the training stage, MLP performance were always close to that of the testing stage. This is the reason why we showed 95%CI, to help the reader understand the steady performance of the model as the number of inputs was reduced. In our opinion, the lack of a strong infection marker allowed the MLP to produce a slight aberration, capturing a faint variable comparing to the count of microbes. Actually, we might consider this a kind of overfitting, as the faint variable captured the noise in the model, which strongly tied with the microbes variable. As a consequence a spurious association within the model arose. A brief explanation was added accordingly in the text.

Comment 4: Acknowledge this limitation more explicitly in the conclusion.

Response 4: a sentence was added in conclusions to address the helping with synthetic resampling

Comment 5: In the abstract section, write: “Contaminated samples represented 3.5% of cases and were predominantly misclassified as negative by the MLP.” Instead of Contaminated samples were 3.5%, and the MLP classified them as negatives.”

Response 5: The criticism was accepted, and the change was made accordingly

Comment 6: Figure 1 and Table 3 are central to the findings. Consider labeling them more clearly with meaningful titles (e.g., "Model Performance with Variable Inputs").

Response 6: We agree with the reviewer and concise and more explicative titles were provided

Comment 7: In the abstract and results, use “specimens” instead of “samples” for medical precision.

Response 7: We changed the words according to the reviewer’s suggestion.

Comment 8: Clarify that “microbes” as a variable refers to flow cytometry counts, not CFU.

Response 8: As specified in the materials and methods paragraph, the microbes variable refers to flow cytometry counts.

Comment 9: Reference [11] (Kass) is historically significant. You may also want to reference more recent critiques of the Kass threshold to support your position.

Response 9: We added a recent reference (2023) discussing the microbial threshold guidelines for UTI where it is stated that microbial thresholds considered clinically relevant remain in dispute. (reference 12)

Comment 10: The conclusion is appropriate, but could benefit from a stronger emphasis on the clinical implications and next steps

Response 10: A sentence has been added in the conclusion paragraph (lines 292-294)

Reviewer 2 Report

Comments and Suggestions for Authors

Dear authors.

Thank you very much for this work.

Some questions:

It would be interesting to cross-reference the data already used with that provided by urinary sediment analysis: do you have such data? Analysis of bacteria, mucus, epithelial cells, casts, etc., also provides valuable information.

Was the analysis performed on an outpatient population? Did they have any underlying disease?

How were these “contaminated samples” defined? only by CFU/ml parameter? What type of patient was it? Pediatric? Elderly? Monomicrobial? Polymicrobial? Maybe figure 2 can be improved.

Given that I assume that this model was used in an ambulatory population, do the authors believe it would apply to a population with chronic renal diseases, pediatric patients, pregnant women, etc.?
What would be the approximate savings this tool would allow in terms of costs?
It would be good to add a table showing, in addition to % NPV, PPV, the number of positive and negative samples, how many of these were in the controlled category, and cross-check it with the identification of the microorganism.

Kind regards

Author Response

Comment 1: It would be interesting to cross-reference the data already used with that provided by urinary sediment analysis: do you have such data? Analysis of bacteria, mucus, epithelial cells, casts, etc., also provides valuable information.

Response 1: This type of data is not available for this analysis because the goal was to use the information that can be obtained most quickly from the sample analysis, especially in an automated manner.

Comment 2: Was the analysis performed on an outpatient population? Did they have any underlying disease?

Response 2: The analysis was performed on both the outpatient and inpatient populations.  We didn’t collect information about possible underlying diseases, because our aim was to investigate the utility of an artificial neural network applied directly to data obtained through routine laboratory analysis, as also mentioned in the previous answer.

Comment 3: How were these “contaminated samples” defined? only by CFU/ml parameter? What type of patient was it? Pediatric? Elderly? Monomicrobial? Polymicrobial? Maybe figure 2 can be improved.

Response 3: The contaminated samples were those with mixed flora or absence of growth upon culture, or scarce colonies on the plate. Patients were adults affected by monomicrobial infection. We have improved the resolution of Figure 2 by increasing the DPI to 600.

Comment 4: Given that I assume that this model was used in an ambulatory population, do the authors believe it would apply to a population with chronic renal diseases, pediatric patients, pregnant women, etc.?

Response 4: Certainly. However, the network would need to be restructured in order to learn new characteristics of the population from which the samples originate.

Comment 5: What would be the approximate savings this tool would allow in terms of costs?
It would be good to add a table showing, in addition to % NPV, PPV, the number of positive and negative samples, how many of these were in the controlled category, and cross-check it with the identification of the microorganism
.

Response 5: We didn’t perform any cost analysis yet, because our PPV% is not as good as we expected. We are going to approach this kind of analysis after improving the sensitivity of the method that can be obtained by analyzing a larger number of samples. The number of positive and contaminated samples in the training and test groups was reported in the Materials and Methods section.

Reviewer 3 Report

Comments and Suggestions for Authors

My Comments and Suggestions for the Authors are given below:

In this study, the authors aim to investigate the integration of an ANN for the diagnosis of Urinary tract infections (UTIs) into the pre-analysis workflow to reduce unnecessary urine cultures. The study does not provide a significant theoretical or methodological contribution to the field. There are many modelling studies with similar approaches in the existing literature.

  1. The artificial neural network modelling used does not provide sufficient depth and contribution on how to integrate it into clinical decision processes. The data used is limited to laboratory outputs only. There is a lack of clinical validation. Therefore, its applicability remains limited.
  2. The study has data imbalance, contamination separation, limited variable analysis, lack of external validity and disconnections from the clinical context.
  3. All data are based on a single hospital and the same analysis device.
  4. The method used is a multilayer perceptron (MLP), and only two variables are used in the model.
  5. The Structure and Training of the Model are Insufficient. Details such as network architecture, number of layers, activation functions and training process are explained superficially. Although the authors discussed some limitations, they did not develop methodological responses to them.

This study addresses a problem of high clinical importance with a practical and applicable solution. But the article makes a limited contribution to the field and contains important methodological and conceptual deficiencies. It is not appropriate to publish the study without resolving the above structural deficiencies and interpretation errors.

Comments on the Quality of English Language

--

Author Response

Comment 1: The artificial neural network modelling used does not provide sufficient depth and contribution on how to integrate it into clinical decision processes. The data used is limited to laboratory outputs only. There is a lack of clinical validation. Therefore, its applicability remains limited.

Response 1: The aim of the study is to identify a strategy for selecting contaminated samples in order to reduce the number of unnecessary urine cultures. There is no diagnostic purpose; therefore, the use of basic information—i.e., data provided directly by the automated analysis—represents a strength rather than a limitation.

Comment 2: The study has data imbalance, contamination separation, limited variable analysis, lack of external validity and disconnections from the clinical context.

Response 2: The data imbalance reflects the actual prevalence of truly positive and contaminated samples in our population. Of course, it can be mitigated with a synthetic data sampling strategy; however, this does not guarantee a real improvement in the MLP's discriminative power. What the study highlights is the limitation of the urinalysis reading, which tends to saturate and flatten the discrimination between samples—a limitation for which no synthetic sampling technique can provide a solution. The only remedy is a laboratory one: diluting the urine. As for the alleged disconnection from the clinical context, it is unclear what the reviewer specifically means by this.

Comment 3: All data are based on a single hospital and the same analysis device.

Response 3: In fact, the test clearly indicates that a limitation of the work is the tendency of urine to flatten the discrimination of samples with a high microbial count, due to the characteristics of this particular instrument.

Comment 4: The method used is a multilayer perceptron (MLP), and only two variables are used in the model.

Response 4: The solution to the XOR logical function problem was also achieved using an MLP with only 2 inputs. Certainly, the problem could also be addressed with logistic regression, but not with the same flexibility and potential for future updates.

Comment 5: The Structure and Training of the Model are Insufficient. Details such as network architecture, number of layers, activation functions and training process are explained superficially. Although the authors discussed some limitations, they did not develop methodological responses to them.

Response 5: Figure 3 and Table 5 contain all the information necessary and sufficient to replicate and verify this model, even using Excel. The model’s limitations do not lie in its architecture or in data imbalance, but rather in the data source itself, as bacterial counts quickly reach saturation, limiting the ability to discriminate between positive and contaminated samples. As already mentioned, the methodological response is to dilute the urine samples.

Comment 6: This study addresses a problem of high clinical importance with a practical and applicable solution. But the article makes a limited contribution to the field and contains important methodological and conceptual deficiencies. It is not appropriate to publish the study without resolving the above structural deficiencies and interpretation errors.

Response 6: it remains unclear what the serious methodological and conceptual flaws affecting the work actually are. It is fair for the reviewer to request that these issues be addressed before publication; however, by remaining vague in their comments and therefore not allowing for an accurate and specific response, their remarks come across as an ideological opposition rather than a methodological critique.